# “Nitrogen Wash-Out” in Non-Hypoxaemic Patients with Spontaneous Pneumothorax: A Narrative Review

**DOI:** 10.3390/jcm12134300

**Published:** 2023-06-27

**Authors:** Erwin Grasmuk-Siegl, Arschang Valipour

**Affiliations:** 1Department of Respiratory and Critical Care Medicine, Klinik Floridsdorf, Brünner Straße 68, 1210 Vienna, Austria; 2Karl-Landsteiner-Institute for Lung Research and Pulmonary Oncology, Health Care Group, Klinik Floridsdorf, Brünner Straße 68, 1210 Vienna, Austria

**Keywords:** pneumothorax, nitrogen wash-out, oxygen therapy

## Abstract

Following current guidelines, spontaneous pneumothorax should be primarily managed with minimal invasive strategies. In real-world clinical practice, oxygen supplementation regardless of the presence or absence of hypoxemia is frequently applied in patients with a pneumothorax, with the intention to enhance the resorption rate of air from the pleural cavity (“nitrogen wash-out theory”). This review provides an overview of the scientific origin of this practice in animal models, and its clinical use in adult and paediatric patients. Clinical studies from PubMed, Embase and Cochrane library were reviewed by the authors using the keywords, “oxygen AND pneumothorax”, “nitrogen washout AND pneumothorax” and “nitrogen AND pneumothorax”, and recommendations from current guidelines were also reviewed by the authors. A selected total of nine clinical studies and three guidelines were included. Though in animal models there appears to be a therapeutic effect of oxygen therapy for the treatment of pneumothorax, clinical data in patient populations mainly stem from retrospective studies, mostly with a small sample size and inadequate study design. We recommend conducting prospective clinical studies with adequate methodology to address the question of whether or not oxygen therapy should be used to treat pneumothorax, regardless of the presence or absence of hypoxemia.

## 1. Introduction

A pneumothorax is an air accumulation within the pleural space and may occur spontaneously (e.g., rupture of an emphysematous bullae), traumatic or iatrogenic (e.g., secondary to transbronchial or transthoracic lung biopsy). Spontaneous pneumothorax can further be categorised into primary or secondary events, depending on whether an underlying pathology of the lung is present (e.g., chronic obstructive pulmonary disease, interstitial lung disease) or not. According to a French study covering the period 2008 to 2011, the incidence of spontaneous pneumothorax in the general population is 22 cases per 100,000 population, with males between 20 and 35 years of age being predominantly affected [1,2].

Since guidelines for the treatment of a spontaneous pneumothorax primarily recommend minimal invasive strategies, such as needle aspiration or watchful-waiting [2,3,4], oxygen therapy in the absence of hypoxemia remains debatable. The administration of oxygen in patients with a pneumothorax should theoretically hasten the re-expansion of the collapsed lung, by washing out nitrogen from the arterial blood and should therefore increase the absorption gradient between blood and air in the pleural cavity. The scientific evidence for this proposed mechanism of action, however, appears to be rather scarce. To our knowledge, this narrative review is the first to systematically discuss the roots of this theory using both a historical perspective and a critical glance of this commonly applied practice in clinical care.

## 2. Methods

We searched PubMed, Embase and Cochrane library by using the keywords, “oxygen AND pneumothorax”, “nitrogen washout AND pneumothorax” and “nitrogen AND pneumothorax”, without restriction regarding the publication year. Both human studies and animal studies were considered eligible. Only studies with available abstracts online were included. If full texts were not available digitally, they were collected in the library of the Medical University of Vienna.

In addition, recommendations from current guidelines by the British Thoracic Society, European Respiratory Society as well as the German Society for Thoracic and Cardiovascular Surgery were reviewed for their references regarding the use of oxygen in non-hypoxemic patients with pneumothorax. 

A selected total of nine studies that were considered relevant are summarised in Table 1.

The didactic structure of this paper is according to the Scale for the Assessment of Narrative Review Articles (SANRA) [5].

**Table 1 jcm-12-04300-t001:** Overview of interventional studies on nitrogen wash-out for the treatment of pneumothorax. Abbreviations: cm^2^/day= square centimetres per day; L/min = litres per minute; na = not applicable; nr = not recorded. Sorted by date of publication.

Author	Year	Study Design	Population	Type of Pneumothorax	n =	Age [Years]	Measurement	Control-Group	Initial Size Oxygen Group	Initial Size Control Group	Resolution Oxygen Group	Resolution Control Group	Oxygen Therapy (Dose and Duration)
Northfield [6]	1971	Longitudinal observation with retrospectively observed control group	human	Spontaneous pneumothorax	22	17–51 (mean 25)	Kirchner´s method; daily X-ray	na	nr	nr	mean resolution rate: 17.9 cm^2^/day	Mean resolution rate: 4.8 cm^2^/day	16 L/min; 9 to 38 h at a time
Chadha [7]	1983	Retrospective chart review	human	Primary-/secondary spontaneous, iatrogenic	8	nr	Kirchner´s method; “periodic” X-ray	na	19% [area of the hemithorax]	na	mean resolution rate: 4.2%/day	na	3 L/min via nasal canula
Cormier [8]	1980	Double-blinded randomised	human	Iatrogenic pneumothorax	46	nr	Serial X-ray	20	na	na	nr	nr	100% via mouthpiece
Hill [9]	1995	Case-control study in animal model	New Zealand rabbits	Iatrogenic pneumothorax	20	nr	Serial X-ray till complete resolution	11	nr	nr	no means reported; complete resolution: 36 h in majority	no means reported; “majority in the room-air group did not show complete resolution before 48 h.”	
England [10]	1998	Dose-finding study with four groups	New Zealand rabbits	Iatrogenic pneumothorax	40	nr	Serial X-ray till complete resolution	na	nr	nr	no means reported; Complete resolution: 30% FiO_2_: 42.90 (±5.97) h, 40% FiO_2_: 35.80 (±4.26) h, 50% FiO_2_: 33.80 (±4.66) h	complete resolution: 61.65 (±12.30) h	30%, 40%, 50%
Zierold [11]	2000	Dose-finding study with three groups	New Zealand rabbits	Iatrogenic pneumothorax	20	nr	X-ray twice daily till complete resolution	na	nr	nr	mean complete resolution: 40% FiO_2_: 71.8 (±22.3) h 60% FiO_2_: 39.4 (±14.2) h	mean complete resolution: 111.2 (±30.8) h	40%, 60%
Shaireen [12]	2014	Retrospective population-based cohort study	human	Primary spontaneous pneumothorax	92	neonates	initial X-ray, Clinical resolution defined as cessation of respiratory distress and discontinuation of oxygen treatment with maintenance of SpO_2_ ≥ 95%.	30	nr	nr	median time to clinical resolution: FiO_2_ < 60%: 12 (5–24) h FiO_2_ ≥ 60%: 12 (8–27) h	median time to clinical resolution: 11 (4–24) h	FiO_2_ < 60% and ≥60% via oxyhood or nasal prong
Clark [13]	2014	Retrospective chart review	human	Primary spontaneous pneumothorax	45	neonates	Serial X-ray	na	nr	nr	median time to clinical resolution: 37 (±27) h	median time to clinical resolution: 20 (±26) h	FiO_2_ 100%
Park [14]	2017	Retrospective chart review	human	Primary spontaneous pneumothorax	175	19.26 (±4.93)	serial X-ray with variable interval	47	percentage of thoracic cavity: 23.32 (±7.00) %	percentage of thoracic cavity: 20.26 (±6.78) %	percentage of thoracic cavity: 4.27 (±1.97) %/day	percentage of thoracic cavity: 2.06 (±0.97) %/day	2–4 L/min via a nasal cannula

## 3. Application of the Nitrogen Wash-Out Theory in Animal Models

The origins of the nitrogen wash-out theory date back to the scientific literature of the 1930s: Henderson Y and Henderson MC wrote in their theoretical paper about absorption of gas from any closed space within the body in 1932 [15]. As environmental air passes freely into the alveolar space, the sum of partial pressures of nitrogen, oxygen, carbon dioxide and water vapor reach the pulmonal capillaries at 760 mmHg (at sea level). During gas exchange in the peripheral tissue, oxygen leaves the blood and carbon-dioxide enters in a ratio depending on the respiratory quotient (CO_2_-removal/O_2_-consumption; normal range: 0.7–1). Thus, the changes in the partial pressure of oxygen are much more pronounced than in carbon dioxide. Since pressures of nitrogen and water vapor remain unchanged, venous blood leaves the capillaries with a reduced sum of 706 mmHg. This gradient between venous blood and environmental air is the driving physical force which leads to the absorption of any gas in closed spaces within the body. Air “trapped” in the pleural cavity follows this gradient of 54 mmHg (environmental pressure minus pressure in venous blood) towards the blood resulting in a subsequent resolution of a pneumothorax over the course of time [15]. The above gradient could be theoretically increased by replacing nitrogen, by breathing highly concentrated oxygen. With the proportion of nitrogen in the sum of partial pressures in arterial blood leaving the pulmonary capillaries decreasing, there is an enhanced diffusion gradient between the pleural cavity and blood [16]. 

In 1935, Fine and colleagues studied the enhanced absorption rate of air injected subcutaneously in rabbits breathing room air or 95% oxygen, and thus laid the foundation for the nitrogen wash-out theory. One hundred cubic centimetres of room-air was injected into the subcutaneous tissues of two separate rabbits. One was placed in a box filled with 95% oxygen, the other one in a box filled with room-air. X-rays were taken right after the procedure as well as in the following 6, 12 and 24 h. After 24 h the X-ray of the rabbit from the oxygen-box not only showed a complete resolution of the subcutaneous air depot, but also a complete absorption of the intestinal gas, compared with the rabbit from the room-air-box [16]. Fine and colleagues extended their findings beyond this point, by demonstrating a potential therapeutic effect of highly concentrated oxygen in a case series of distended small intestines [17]. According to the theory proposed, the accelerated absorption should continue until the partial pressures equalise, which according to Piiper J. (1965) is the case when 60% of the volume has been absorbed [18]. Comroe wrote in his textbook on lung physiology about the therapeutic usage of oxygen in patients with pneumothorax based on the above-mentioned observations [19].

In the mid-nineties, Hill and colleagues evaluated oxygen therapy in an animal-model: With needles and syringes, air was injected in the pleural space of 20 New Zealand rabbits [9]. Nine of these rabbits were placed in a cage with room air, whilst eleven were placed in a cage with high oxygen concentration (60%). The resolution rate of pneumothorax was observed via X-ray, three times a day. After 36 h the pneumothorax of rabbits in the high-oxygen cage had completely resolved. Although two rabbits in the room-air cage had complete resolution within 24 h, the other seven animals in this cage had their pneumothorax resolved after 48 h. The report states that the oxygen concentration in the pleural cavity was measured but does not provide respective values. Similarly, there is no data on nitrogen concentrations in the publication. On the basis of their findings, however, the same research group conducted a subsequent study with forty rabbits, observing a significant dose response of pneumothorax resolution and administered oxygen levels (oxygen levels 30–50%) [10]. 

It must be acknowledged, however, that the pneumothorax was induced artificially by injuring the pleura. Thus, the findings may not be applicable to the nature and pathophysiology of a spontaneous pneumothorax in humans.

Another report by Zierold et al. similar assessed the concept of nitrogen wash-out in 20 rabbits with artificially induced pneumothorax by thoracoscopically guided visceral pleural punctures [11]. Six rabbits were subsequently placed in cages filled with room air and seven rabbits each were placed in cages with oxygen levels of 40% and 60%. X-rays were performed twice a day until the pneumothorax resolved completely. The time to complete resolution was 111.2 (±30.8) h, 71.8 (±22.3) h, and 39.4 (±14.2) h on room air, 40%, and in the 60% group, respectively [11]. The authors reproduced previous findings by demonstrating a beneficial and dose dependent effect of oxygen on the pneumothorax absorption rate. 

Shih and colleagues [20] investigated the potential downside of high oxygen concentrations in rabbits, i.e., the inflammatory response of lung tissue exposed to hyperoxia. Twenty-eight animals received an artificially induced pneumothorax and were treated with different oxygen concentrations. Broncho-Alveolar Lavages were obtained after full resolution of the pneumothorax, measuring intrapulmonary levels of IL-1β and IL-8, as a surrogate of acute inflammatory lung injury. Among rabbits receiving FiO_2_ levels greater than 80%, significantly higher levels of IL-1β were detected, leading to the authors’ recommendation that lower oxygen concentrations (60%) may be safer to use in that specific context.

## 4. Early Clinical Studies of Applying the “Nitrogen Wash-Out” Theory for the Management of Pneumothorax

In 1971, Northfield and colleagues [6] published a case control study with a total 22 patients suffering a spontaneous pneumothorax. The study lacks more in-depth description of patient characteristics beyond age (mean 34 yrs). Furthermore, only one woman was included. Two patient-groups were studied: A retrospective cohort of twelve patients with a spontaneous pneumothorax not receiving oxygen therapy, which were prescribed bed rest only. Furthermore, 10 patients were prospectively studied and received mask oxygen therapy “repeatedly” with a flow of 16 L/min for 9 to 38 h at a time. The timespan without oxygen therapy from each of these patients was statistically evaluated as interindividual control. The size of pneumothorax was assessed via daily chest X-rays. The pictures were superimposed on graduated graph paper and reported by two separate “observers”. During oxygen therapy the absorption rate increased by a factor of 3.7 to a mean value of 17.9 cm^2^/24 h, compared to the historic control group. A larger effect was observed in patients with an initial pneumothorax size of more than 30% of the pleural cavity. 

Twelve years later, Chadha and colleagues published a case series with eight patients that experienced a pneumothorax [7]. The genesis of pneumothorax in this study collective was rather heterogeneous: Two primary spontaneous pneumothorax, three patients with secondary spontaneous pneumothorax with underlying lung diseases, and three iatrogenic pneumothoraxes were included. All patients received high concentrations of oxygen via a rebreathing mask. The size of the pneumothorax on the posterior-anterior X-ray was expressed as a percentage from the hemithorax. The resolution rates of pneumothorax were compared to previous studies, which suggested an absorption rate of 1.25% per day by Kircher and Swartzel [21] and a previously published 21 days for full resolution of a pneumothorax greater than 25% of a hemithorax, solely treated with bed rest by Hickok and Ballenger [22]. Six of the patients studied in the report by Chadha et al., with a mean expansion of the pneumothorax below 30% of the hemithorax, showed a mean resolution rate of 4.2% per day, and therefore faster than compared to room air. After one week of oxygen treatment, the pneumothorax of all six patients was resolved completely. Two patients with expansion of more than 30% needed a chest tube and were therefore excluded from the analysis.

In 1980, Cormier [8] and colleagues performed a double-blind randomised trial on prophylactic oxygen administration during ultrasound guided transthoracic needle-biopsies. Twenty-six patients were randomised in the room air group and 20 in the oxygen group receiving 100% oxygen. The authors cite the report by Dale and Rahn [23] to “justify” their approach. The latter report demonstrated that the absorption rate of an oxygen-filled lung is 62 times faster than for a nitrogen filled one. Even if diffusion through the pleural barrier is slower than through the alveolar wall, the ratio between oxygen and nitrogen may remain the same. Patient characteristics in the report by Cormier et al., especially smoking history and age did not differ significantly between the groups. Forty-two percent of patients in the room air group and 20% in the oxygen group suffered an iatrogenic pneumothorax. All pneumothorax in the oxygen group appeared within the first ten minutes after biopsy. The main diagnosis made by biopsy in both groups were cancer and tuberculosis. Expectedly, the depth of the respective target lesion within the lung appeared to correlate with the occurrence of a pneumothorax. In the room air group, two patients with the lesion adjacent to the pleura and four with a lesion in a maximum depth of two centimetres suffered a pneumothorax. In the oxygen group, none of the patients with lesions next to the pleura and only one with a lesion in a maximum depth of two centimetres suffered a pneumothorax. The above findings must be placed into the context of more recent advancements in imaging and puncture techniques. In fact, a recent survey on the safety of ultrasound guided transthoracic needle biopsy in 762 patients showed an incidence for iatrogenic pneumothorax of 0.79% [24]. None of these patients required chest-tube drainage. 

Noteworthy, the studies from Northfield et al. and Chadha et al. used a method for calculating the pneumothorax size which was published by Kircher and Swartzel in 1954 (see Figure 2 in Park CB et al. [14]) and was based on mathematical models obtained from a small clinical sample size. The resolution rate of pneumothorax in room air was estimated at 1.25% per day [14,21]. The Kircher´s method, however, no longer finds any clinical application nowadays. In 1995 the Collin´s method was published, which provides more precise estimates of the pneumothorax size in the erect posteroanterior radiographs compared to CT scans (see Figure 1 in Park CB et al. [14]) [25]. The resolution rate of conservatively treated spontaneous pneumothorax without oxygen-therapy was calculated at 2.2% per day according to the Collin´s method [26].

## 5. 21st Century Data

The retrospective study by Park and colleagues from 2017 provides the most recent data on oxygen therapy for pneumothorax resolution [14]. The records of 175 patients with primary spontaneous pneumothorax from an eleven-year period were retrospectively reviewed. One-hundred twenty-eight patients received bed rest plus continuous oxygen via a nasal cannula (2–4 L/min) on admission (oxygen group). Forty-seven outpatients on room air served as a control group. Patients with a primary spontaneous pneumothorax beneath 40 years of age were included resulting in a mean age in both study groups of 19 years. 

The mean initial size of the pneumothorax—measured via the Collin´s method—was slightly larger in the oxygen group (20.26 ± 6.78 % (room air) vs. 23.32 ± 7.00% (oxygen)); percentage of thoracic cavity). The time between the follow-up X-rays differed significantly between the control group of outpatients and the in-patient oxygen group (4.2 ± 2.3 days (room air) vs. 2.5 ± 1.1 days (oxygen)). The resolution rate (%/day = [initial size (%) − last size (%)]/time interval (days)) was subsequently calculated on the basis of initial pneumothorax size and estimated daily improvements based on follow-up X-rays. According to this approach there was a significantly higher resolution rate in the oxygen group (2.06 ± 0.97%/day (room air) vs. 4.27 ± 1.97%/day (oxygen)). The last measured pneumothorax size between groups, however, was less than one percent and did not differ significantly. Furthermore, the report suffers from very important limitations. First, fourteen patients in the oxygen group suffered an increase in pneumothorax size and were excluded from statistical analysis. Second, fifteen patients had two episodes of pneumothorax during the study period, with 13 of these included repeatedly into the oxygen group. Both limitations result in an important bias towards the intervention group of patients receiving oxygen therapy.

## 6. Neonatal and Paediatric Patients

Whilst data on the effect of nitrogen washout in adult patients with pneumothorax may indicate some benefits, the evidence in neonatal and paediatric patients speaks strongly against the usage of oxygen for accelerating the rate of pneumothorax absorption. In fact, in a cohort study of 92 in-term infants suffering from a spontaneous pneumothorax, there was no evidence of faster resolution of pneumothorax in those receiving oxygen compared to a control group treated with room air [12]. Along these observations, a retrospective chart review of a small sample of neonates with spontaneous pneumothorax compared “conventional therapy” (n = 19) and oxygen supplementation (n = 26) and did not observe an enhanced absorption rate in the latter [13]. These findings were recently summarised in a review by Wilson and colleagues, with no recommendations being given for oxygen therapy in the management of pneumothorax in adolescent patients [27].

## 7. Pathophysiological Effects of Hyperoxia

In addition to the above-mentioned limitations in the methodology of clinical studies assessing the use of oxygen therapy in the treatment of pneumothorax, it has to be kept in mind that supplementing oxygen in normoxaemic patients may lead to unfavourable effects. First and foremost, hyperoxia may increase the production of reactive oxygen species (ROS). ROS are by-products of mitochondrial metabolism. During adenosine triphosphate production via oxidative phosphorylation, ROS arise with increasing oxygen tension. These by-products are substrates of intracellular signalling pathways and subsequently lead, among other effects, to vasoconstriction [28]. Therefore, supra-physiological levels of oxygen in the blood may reduce overall oxygen delivery to peripheral vasculatures, such as coronary, renal or cerebral [29]. In fact, during hyperoxia, cerebral blood flow— for example—is reduced up to a third [30]. Besides vasoconstriction, ROS in high concentrations is able to induce cell apoptosis and cellular damage by initiating inflammatory pathways in the alveolar epithelium [28]. 

Overall, the recommendation of the British Thoracic Society [28] summarise the physiological risks of hyperoxia due to supplemental oxygen therapy as follows:Worsening V/Q mismatchAbsorption atelectasisCoronary and cerebral vasoconstrictionReduced cardiac outputDamage from oxygen free radicalsIncreased systemic vascular resistance

The potential negative consequences of hyperoxia have also been demonstrated in a number of clinical settings. As such, patients after successful resuscitation may have a poor neurological outcome (OR 1.37 [95% CI 1.01 to 1.86]) as well as overall higher mortality (OR 1.32 [95% CI 1.11 to 1.57]) following hyperoxia, according to a meta-analysis [31]. Similarly, the aforementioned vasoconstriction caused by ROS may worsen outcomes following a myocardial infarction. The prospective, randomised and controlled AVOID (Air Verses Oxygen In myocarDial infarction) trial questioned the routine supplemental oxygen therapy (8 L/min) in patients with acute ST-elevation myocardial infarction and found adverse outcomes associated with usage of oxygen in normoxaemic patients. Oxygen supplementation in the absence of hypoxemia was associated with a higher risk of recurrent myocardial infarction (5.5% versus 0.9%; *p* = 0.006) [32]. 

Finally, breathing highly concentrated oxygen may result in absorption atelectasis. When pure oxygen is administered, the pressure gradient between alveolar space and mixed venous blood in pulmonary vasculature exceeds the physiological amount by replacing nitrogen with oxygen. Subsequently, gas shifts towards the blood—similar to the pleural cavity—causing partial alveolar collapse and therefore an unfavourable ventilation/perfusion balance through formation of resorption atelectasis [30,33].

## 8. Current Recommendations

In a Task Force statement by the European Respiratory Society for the treatment of primary spontaneous pneumothorax, application of oxygen does not find any mentioning [3].

The guideline for emergency oxygen use in adult patients by the British Thoracic Society (2008) recommend the usage of a reservoir mask at 10–15 L/min with a target peripheral oxygen saturation of 100% to “accelerate clearance of pneumothorax if drainage is not required” [29]. In its separate guideline for the management of spontaneous pneumothorax (2010), the British Thoracic Society specified the therapeutic effect with a quadrupling resorption rate and cited the above outlined early clinical studies by Northfield et al. [6] and Chadha et al. [7] as the underlying references. The 2018 guidelines for the management of spontaneous pneumothorax by the German Society for Thoracic and Cardiovascular Surgery argues on the basis of the same data but acknowledges at the same time that oxygen therapy as a single intervention may not be sufficient [2].

A clinical guideline published in 2007 by Currie et al. recommends titrating patients with a pneumothorax to a peripheral oxygen saturation above 92%, using an oxygen concentration above 28% FiO_2_, without citing specific sources of information for this recommendation [34].

A more recent review-article questions the therapeutic effects of oxygen therapy in the management of a pneumothorax, particularly in patients with underlying lung diseases (e.g., COPD) referring to the potential harmful effects of hyperoxemia in specific populations, thus limiting the net benefits of the nitrogen-washout mechanism to resolve a pneumothorax [35].

## 9. Future Studies

Where to go from here? There is a need for an adequately designed, powered, randomised, controlled clinical trial in normoxaemic patients with a pneumothorax. Ideally, the study should include a homogenous patient population, such as patients with a spontaneous pneumothorax. It may also be conceivable to include patients with a pneumothorax following transbronchial lung biopsy. In that case, a subgroup analysis would need to be defined a priori. Titration of oxygen supplementation based on an upper threshold of SpO_2_ (e.g., 98%) should be performed. Alternatively, all patients included should receive a fixed oxygen concentration irrespective of the underlying SpO_2_ level. In addition, the method of oxygen therapy would need to be standardised (nasal cannula vs. face mask). Both the method (X-ray vs. low-dose CT scans vs. lung ultrasound) and duration of follow-up should be predefined, as well as supportive measures, such as bed rest, analgesics, and/or need for thromboprophylaxis. Finally, it remains to be determined whether the control group would need to receive a sham-intervention, such as high-flow room air supplementation through a nasal cannula. 

## 10. Summary

According to previous animal models there appears to be a therapeutic effect of oxygen therapy for the treatment of pneumothorax, based upon the nitrogen wash-out theory. 

The transfer from the findings in the animal model to humans, however, may be less clear. Clinical data mainly stem from retrospective studies, mostly with a small sample size. Most of these studies, however, appear not to meet the scientific requirements to continue to serve as a reference for clinical recommendations. Future studies with adequate design and cohort size are warranted, to address the question whether or not to apply oxygen therapy in non-hypoxemic patients in the management of pneumothorax.

## Data Availability

Not applicable.

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
