# Peer review of "“Nitrogen Wash-Out” in Non-Hypoxaemic Patients with Spontaneous Pneumothorax: A Narrative Review"

_jcm, 2023, doi:10.3390/jcm12134300_

Round 1
Reviewer 1 Report (Previous Reviewer 1)
The authors responded to reviewers' comments and made improvements to the paper accordingly. I believe that in the current version the review is quite comprehensive and interesting for the readers of the journal.
Author Response
Answer to Reviewer 1:
We thank the reviewer for his interest in our manuscript and the recommendation for acceptance in the Journal of Clinical Medicine.
Reviewer 2 Report (New Reviewer)
The authors of this review try to provide an overview of oxygen therapy in the absence of hypoxemia hasten the re-expansion of the collapsed lung, by washing out nitrogen from the arterial blood. Although the topic of this study is interesting, I have several concerns regarding this manuscript. Below you can see my detailed comments:
1) the review is comprehensive and somehow historical, but unfortunately I can’t see the novelty of this review. The theory of nitrogen-wash out is well known. Please rephrase the objectives of this review and why it is important?.
1) The introduction: it is very short; perhaps you can add a brief definition of what pneumothorax is and its different types.
2) In the following section “Early clinical studies of applying the “nitrogen wash-out” theory for the management of pneumothorax”, most of the used references are very old. I think these studies need to be updated.
- The methods: Please change from table 1 to “Table 1”. Also, I noticed multiple punctuation errors in all the manuscript that need to be edited.
-Future studies section: need to be rephrased. Some grammar and punctuation errors need to be fixed.
-In line 336: BTS abbreviation is written without previous illustration. Please, revise all other abbreviations.
Author Response
Suggestions Reviewer 2:
- The review is comprehensive and somehow historical, but unfortunately, I can’t see the novelty of this review. The theory of nitrogen-wash out is well known. Please rephrase the objectives of this review and why it is important.
Authors response: We acknowledge that, at first glance, the topic of oxygen treatment for pneumothoraces may not immediately draw (scientific) attention. However, after careful review of the available literature, we believe that this review may be of specific relevance as it is the first to systematically question the underlying evidence base for this commonly applied therapeutic option in clinical practice. As the reviewer has correctly identified, most of the scientific data related to the nitrogen washout theory dates back to the 60s and 70s and thus in our opinion is outdated. More recent data remains controversial and there is an obvious lack of a prospective randomised controlled trial on this topic. Thus, we hope that the reviewers considers acceptance of this manuscript for what it is, i.e. a critical assessment of the paucity of data regarding a frequently applied technique across large parts of the world.
- The introduction: it is very short; perhaps you can add a brief definition of what pneumothorax is and its different types.
Authors response: We believe that readers of our article belong to a medical group for which the clinical picture of pneumothorax is part of their general knowledge, in particular because we are dealing with a clinical topic. However, in order to meet the reviewer's suggestion, we added a short paragraph on the basic definition and incidence of pneumothoraces.
- In the following section “Early clinical studies of applying the “nitrogen wash-out” theory for the management of pneumothorax”, most of the used references are very old. I think these studies need to be updated.
Authors response: As mentioned earlier we are aiming to give the readers an understanding of the evidence base of the nitrogen wash-out theory. Northfield et al (1971) and Chadha et al (1983) are the first two studies, applying the theory in clinical practice. Between the mid-eightees and the study by Park et al in 2017 no relevant investigations were published in this field. Northfield and Chadha´s studies form the basis for recomandations by the German Society for Thoracic and Cardiovascular Surgery and the British Thoracic Society. We therefore felt that is important to highlight the weak scientific evidence based for those recommendations, based on outdated publications and lack of high-quality data.
Comments on the Quality of English Language
Authors response: We appreciate the reviewers comment and have applied both a gramar and punctuation check to the manuscript.
This manuscript is a resubmission of an earlier submission. The following is a list of the peer review reports and author responses from that submission.
Round 1
Reviewer 1 Report
Comments to Authors
In the present manuscript, Dr. Erwin Grasmuk-Siegl and Arschang Valipour provide a narrative review of the nitrogen wash-out theory and the role of the administration of oxygen in normoxaemic patients with spontaneous pneumothorax.
The hypothetical basis is that oxygen therapy reduces the partial pressure of nitrogen in the alveolus compared with the pleural cavity, and a diffusion gradient for nitrogen accelerates resolution.
A total of nine clinical studies were selected as they were considered relevant.
The article is well written, the only concern is the interest it can arouse in the readers of the journal.
The main evidence is that there is insufficient evidence to recommend oxygen therapy in non-hypoxemic patients with spontaneous pneumothorax.
However, this could be a starting point for conducting further well-controlled prospective studies.
I would like the authors to consider the following suggestions:
1. The word "Evidence" appears twice in the title when there is very little evidence on this topic, as stated by the authors themselves. Probably a title more corresponding to the content of the paper could be the following:
“Nitrogen Wash-Out” in Non-Hypoxaemic Patients with Spontaneous Pneumothoraces: A historical (or narrative) Review
2. Abstract: “Administering highly concentrated oxygen in normoxaemic patients is a well-established method to enhance resorption rate of air from the pleural cavity addressing the nitrogen wash-out theory”. I disagree with this sentence, since the main guidelines recommend the use of oxygen in hypoxemic patients, with a SpO2 target usually between 94-98% (lower in COPD patients).
3. The authors should emphasize that studies on the use of hyperoxic management have found various adverse effects. Oxygen supplementation induces the release of reactive oxygen species, which inhibit the generation of vasodilators (prostaglandins, nitric oxide). Subsequent vasoconstriction can decrease regional perfusion and lead to chest pain, cough, headache, etc.
- Lumb AB, Walton LJ. Perioperative oxygen toxicity.Anesthesiol Clin 2012;30:591-605.
- Iscoe S, Beasley R, Fisher JA. Supplementary oxygen fornonhypoxemic patients: O2 much of a good thing? CritCare 2011;15:305
Even if some studies showed that the resolution rate of PSP was increased with oxygen supplementation, the routine use of oxygen therapy in patients with small pneumothoraces should be considered carefully taking into consideration current concerns about adverse outcomes of hyperoxia.
- Park CB, Moon MH, Jeon HW, ChoDG, Song SW, Won YD, Kim YH, Kim YD, Jeong SC, KimKS, Choi SY. Does oxygen therapy increase the resolution rate of primary spontaneous pneumothorax? J Thorac Dis2017;9(12):5239-5243.
Reviewer 2 Report
I am grateful to the authors for the opportunity to review the paper: "Evidence of "Nitrogen Wash-Out" in Non-Hypoxaemic Patients with Spontaneous Pneumothoraces: An Evidence-based Review".
This work appears to be meaningful and holistic. The authors analyzed a large amount of published research on this topic. However, I have a few comments that may be useful to improve the quality of the work:
1. English language and style should be slightly edited: e.g. in Abstract section there is “appear not to not meet” which may be a bit confusing; “debateable” is spelled in the way which is archaic and the modern spelling is “debatable”; the word “pneumothoraces”, in my opinion, is commonly used in singular form, or the term “inflammatory reaction” should be replaced with “inflammatory response” to my mind. In some places an adjective should be put in the form of an adverb, etc.
2. In section 3 "Application of the nitrogen wash-out theory in animal models", it is advisable to present a graphic drawing or diagram so that the reader could clearly and visually see the physiological rationale underlying the nitrogen wash-out theory. I would recommend to include the entire trajectory in the diagram: from the alveolar gas equation to the pressure gradient between venous blood and air in closed spaces inside the body under standard conditions. Also in this diagram it would be also appropriate to show the physiological changes and the resulting values of pressure gradients during oxygen therapy. In any case, in the review it is desirable to present the physiological and pathophysiological aspects of the problem discussed in the literature in a larger volume, which will increase the value of the review in terms of its theoretical significance.
3. The presented review does not clearly present the current paradigm for doctor decision making in clinical practice. There are objective reasons for this, including the lack of evidence base. However, it would be appropriate if the authors described in more detail the safety issues and risks of the proposed approach, first of all, oxygen-associated lung damage (inflammation, resorption atelectasis formation, etc.). It is also important that the authors specify the cohorts of patients who could potentially benefit from oxygen therapy for spontaneous pneumothorax, the optimal therapy regimen (FiO2, duration of therapy, supporting techniques). It is advisable for the authors to formulate possible ways to overcome the current controversy and outlined plans for further clinical studies that would answer the question on the place of oxygen therapy in the treatment of spontaneous pneumothorax.
In summary: This review highlights an intriguing and understudied topic in modern clinical medicine. However, some additions (described above) are needed in order for the reader to form a clear understanding of the current state of the problem, as well as for the clinician to be able to make a well-considered decision in real clinical practice.